# Application of a Novel Algorithm for Expanding Newborn Screening for Inherited Metabolic Disorders across Europe

**DOI:** 10.3390/ijns8010020

**Published:** 2022-03-15

**Authors:** Simon A. Jones, David Cheillan, Anupam Chakrapani, Heather J. Church, Simon Heales, Teresa H. Y. Wu, Georgina Morton, Patricia Roberts, Erica F. Sluys, Alberto Burlina

**Affiliations:** 1Willink Biochemical Genetics Unit, Manchester Centre for Genomic Medicine, Manchester University NHS Foundation Trust, St Mary’s Hospital, Oxford Road, Manchester M13 9WL, UK; simon.jones@mft.nhs.uk (S.A.J.); heather.church@mft.nhs.uk (H.J.C.); hoiyee.wu@mft.nhs.uk (T.H.Y.W.); 2Service Biochimie et Biologie Moléculaire, Groupement Hospitalier Est, Hospices Civils de Lyon, 69002 Lyon, France; david.cheillan@chu-lyon.fr; 3Department of Metabolic Medicine, Great Ormond Street Hospital NHS Foundation Trust, London WC1N 3JH, UK; anupam.chakrapani@gosh.nhs.uk; 4Neurometabolic Unit, University College London Hospitals NHS Foundation Trust and Enzymes Laboratory, Great Ormond Street Hospital NHS Foundation Trust, London WC1N 3JH, UK; simon.heales@gosh.nhs.uk; 5ArchAngel MLD Trust, Registered Charity No. 1157825, 59 Warwick Square, London SW1V 2AL, UK; georginamorton@archangel.org.uk (G.M.); patroberts.nbs@archangel.org.uk (P.R.); 6Helvet Health, Ruelle de la Muraz 4, 1260 Nyon, Switzerland; e.sluys@helvet-group.com; 7Division of Inherited Metabolic Diseases, Reference Centre Expanded Newborn Screening, University Hospital Padova, 35128 Padova, Italy

**Keywords:** newborn screening (NBS), inherited metabolic disorder (IMD), public health, genetics, congenital disorder, lysosomal storage disorder (LSD), inborn errors of metabolism, rare diseases, methodology, algorithm

## Abstract

Inherited metabolic disorders (IMDs) are mostly rare, have overlapping symptoms, and can be devastating and progressive. However, in many disorders, early intervention can improve long-term outcomes, and newborn screening (NBS) programmes can reduce caregiver stress in the journey to diagnosis and allow patients to receive early, and potentially pre-symptomatic, treatment. Across Europe there are vast discrepancies in the number of IMDs that are screened for and there is an imminent opportunity to accelerate the expansion of evidence-based screening programmes and reduce the disparities in screening programmes across Europe. A comprehensive list of IMDs was created for analysis. A novel NBS evaluation algorithm, described by Burlina et al. in 2021, was used to assess and prioritise IMDs for inclusion on expanded NBS programmes across Europe. Forty-eight IMDs, of which twenty-one were lysosomal storage disorders (LSDs), were identified and assessed with the novel NBS evaluation algorithm. Thirty-five disorders most strongly fulfil the Wilson and Jungner classic screening principles and should be considered for inclusion in NBS programmes across Europe. The recommended disorders should be evaluated at the national level to assess the economic, societal, and political aspects of potential screening programmes.

## 1. Introduction

Inherited metabolic disorders (IMDs) are a large class of rare genetic disorders. IMDs are defined as any primary genetic condition in which alteration of a biochemical pathway is intrinsic to specific biochemical, clinical and/or pathophysiological features [1]. Accurate and timely diagnosis is essential for patients with IMDs, because for many IMDs, treatment (including diet) is available that may improve outcomes. However, many patients face difficulty in obtaining an accurate and timely diagnosis because of the number of rare genetic disorders and the heterogeneity of symptoms and phenotypes [2]. It is possible to carry out widespread, routine screening of many IMDs using dried blood spot (DBS) tests. Furthermore, tandem mass spectrometry (MS/MS) and liquid chromatography-tandem mass spectrometry (LC–MS/MS) have drastically advanced screening capabilities. One DBS can now be analysed for an increasing number of disorders, allowing for the expansion of newborn screening (NBS) programmes [3,4]. 

In the United States (US), the Recommended Uniform Screening Panel (RUSP) has allowed for expansion and standardisation of NBS across states. While in Europe there are some countries screening newborns for over 20 disorders—such as Italy, Hungary, or Austria—other countries screen newborns for as few as two disorders [4]. In 2016 in Italy, legislation was passed that every child born should have the right to be screened for almost 40 IMDs for which there is a viable treatment [5]. Following this bold legislation, in 2019, the European Union (EU) Parliament welcomed the introduction of the “Italian model” and a discussion on how this model might be adopted by all EU member states. Yet, despite the willingness to come together across Europe and harmonise NBS programmes [4,6,7], great disparities exist within the EU and there is no uniform approach for expanding NBS.

With the desire to pave the way forward for evidence-based expansion of NBS programmes, a new approach to objectively evaluate and prioritise IMDs for inclusion in NBS programmes was recently proposed in the form of an algorithm [8]. This algorithm was developed based on the Wilson and Jungner classic screening principles [9]. With the NBS evaluation algorithm, it is possible to prioritise disorders for inclusion on screening programmes, utilising objective and measurable criteria. Individual countries could then strategically evaluate prioritised disorders for inclusion in their NBS programmes based on local economic, societal, and political considerations. The algorithm is intended to offer an objective and standardised tool to evaluate disorders for inclusion on NBS programmes.

The objective of this work is to evaluate and rank a comprehensive list of IMDs using the NBS evaluation algorithm, as described in a previous paper by the authors [8].

## 2. Methods

### 2.1. Identification of Disorders for Analysis

A three-step process was used to select which IMDs would be analysed with the NBS evaluation algorithm (see Figure 1). First, we identified the 84 disorders that were initially considered by the US working group to develop the RUSP in Watson MS et al. [10]. Next, we used the Genetic and Rare Diseases (GARD) Database to validate sixty-seven IMDs. Lastly, 48 IMDs were selected for analysis, based on the following three criteria to advantage disorders that are already widely screened for or that have previously been recommended for screening: (1) The disorder is included in the US RUSP Core Conditions or Secondary Conditions list. (2) The disorder is screened for in the following eight countries, that have a similar healthcare expenditure per capita and who screen for over six IMDs: Germany, Sweden, Austria, Australia, Iceland, New Zealand, Italy, and Portugal [11]. (3) The disorder was recommended for screening in one of three peer-reviewed publications [7,12,13] or by the EU Network of Experts on NBS [6]. 

### 2.2. Assessment of Inherited Metabolic Disorders

The NBS evaluation algorithm [8] was used to assess, score, and rank the 48 IMDs. This algorithm is built based on the Wilson and Jungner classic screening principles [9], and consists of three pillars, Condition, Screening and Treatment. Each pillar contains specific weighted criteria to evaluate disorders; for Condition a maximum of 6 points, for Screening a maximum of 3 points, and for Treatment a maximum of 4 points can be attributed, see Figure 2. Each IMD was analysed using the currently available scientific evidence, including peer-reviewed publications, published databases, and national or international screening databases.

Condition: Information on the natural history and frequency of each IMD was gathered from the following references, sequentially checked in June 2021: GARD database, Orphanet portal, MedlinePlus, and PubMed for relevant peer-reviewed publications. For the frequency of the disorder, European birth prevalence or incidence was used preferentially, and worldwide or US data were used as an alternative when European evidence was lacking (see Table A1 in Appendix A).

Screening: Any disorder that is included in a public DBS NBS programme, or that has a registered Conformité Européenne (CE) marked or Food and Drug Administration (FDA) approved DBS assay was assigned two points for the *Screening* “Availability” category. For disorders where this was not the case, further research was performed using PubMed to find any relevant published evidence of a DBS test in development. PubMed was also used to find performance data of the DBS tests, determining if the DBS test had a low false-positive rate by itself or if additional confirmatory strategies were required and available to improve screening performance, such as second-tier enzyme activity tests performed on the same blood spot or multivariate pattern recognition software.

Treatment: A stepwise approach was used to assess the *Treatment* “Availability” category. First, the European Medicines Agency (EMA) website was used to search for approved treatments. If no approved treatment was found, a search was performed on the ClinicalTrials.gov database to identify investigational treatments in phase III development. Peer-reviewed publications were identified which documented other available treatment interventions such as diet, hematopoietic stem cell transplantation (HSCT), or bone marrow transplant (BMT). In this paper, “treatment strategy” includes both EMA-approved and in development treatments, and treatment interventions (such as diet, BMT or HSCT). Only the highest-scoring treatment strategy was used to assess the *Treatment* “Outcomes” category. For the pre-symptomatic initiation of treatment criterion, no score was given if there was no available clinical data.

Total scores were obtained by adding up the sub-scores for each pillar of the algorithm. The IMDs were then ranked based on their total scores (see Table 1).

## 3. Results

### 3.1. Characteristics of IMDs Identified for Analysis

Forty-eight IMDs were selected for evaluation, as described in the Methods section. Table A1 in Appendix A shows some important characteristics of these disorders.

Twenty-one are lysosomal storage disorders (LSD), eight are disorders of organic acid metabolism (DOAM), seven are disorders of amino acid metabolism (DAAM), nine are disorders of fatty acid metabolism (DFAM), three disorders are classified as Other;Nine disorders had a frequency greater than or equal to 1 in 50,000; 10 disorders had a frequency between 1 in 50,000 and 1 in 100,000; seven disorders had a frequency between 1 in 100,000 and 1 in 150,000; eight disorders had a frequency between 1 in 150,000 and 1 in 250,000; 14 disorders had a frequency less than 1 in 250,000;Four disorders are screened for in over 20 European countries, 15 disorders are screened for in 11 to 20 European countries, nine disorders are screened for in at least one, but fewer than 10 European countries, and 17 disorders are not screened for in the European countries covered in Castineras DE et al. 2019 [7].

### 3.2. Scoring and Ranking of IMDs

The 48 IMDs were assessed with the NBS evaluation algorithm and a score was attributed. Table 1 presents the full list of scored disorders ranked by the highest to lowest score (range 0 to 13 points). Burlina et al. proposed a cut-off of 8.5 points and above, based on the validation of the NBS evaluation algorithm with disorders that are already screened for in the United Kingdom NBS screening programme [8]. Using the score of 8.5 as a threshold, there are 35 disorders that most strongly fulfil the Wilson and Jungner classic screening principles and are recommended as candidates for inclusion in NBS programmes across Europe [9]. 

The highest scoring disorders were carnitine uptake defect/carnitine transport defect (CUD), with a score of 12.5 out of 13, and severe combined immunodeficiency (SCID) with a score of 12. Seven disorders scored 11.5 or 11 points, 14 disorders scored 10.5 or 10 points, nine disorders scored 9.5 or 9 points, and three disorders scored 8.5 points. The remaining 13 disorders scored between 1 and 8 points. Of the 20 LSDs analysed with the NBS evaluation algorithm, eight had a score of 8.5 and higher: Pompe disease, Gaucher disease, lysosomal acid lipase deficiency (LAL-D), metachromatic leukodystrophy (MLD), mucopolysaccharidosis type I (MPS I), Krabbe disease, Batten disease (CLN2), and Niemann Pick A/B (ASM deficiency).

With the NBS evaluation algorithm, it is possible to look separately at each of the three pillars and compare different disorders. Looking at all 48 IMDs assessed, in the pillar *Condition*, 77% of disorders (37/48) received points for “all forms of the disorder are asymptomatic for the first few weeks of life” (see Table A2 in Appendix A). For the pillar *Screening*, 31/48 disorders have a “DBS test available in use”, and 26 of these 31 disorders have a DBS test with a “low false-positive rate or a high positive-predictive value (PPV)” (see Table A3 in Appendix A). For the pillar *Treatment*, 21/48 disorders have an available EMA-approved treatment and 65% of disorders (31/48) have a treatment strategy (either an EMA-approved treatment or a treatment intervention) that results in better outcomes if initiated pre-symptomatically (see Table A4 in Appendix A). 

It is also interesting to compare the three pillars of the NBS evaluation algorithm for the 35 top-ranked disorders, those scoring ≥8.5 points. For the pillar *Condition*, all 35 top-ranked disorders have a rapidly progressing form and all but one disorder, PKU, can be fatal by adolescence (see Table A2 in Appendix A). Looking at the pillar *Screening*, of the 35 top-ranked disorders, 33 have a DBS test that is available and in use, and of these 33 DBS tests, 25 have a “low false-positive rate or a high PPV” (see Table A3 in Appendix A). For the pillar *Treatment*, 97% (34/35 disorders) have a treatment strategy available, either an EMA-approved treatment (14/34 disorders) or a treatment intervention (20/34 disorders) (see Table A4 in Appendix A). One disorder, Niemann Pick A/B (ASM deficiency), has a treatment in late-stage development. Importantly, 60% (21/35) of the top-ranked disorders have a treatment strategy available that changes the prognosis for all forms (mild to severe) of the disorder. Alternatively, none of the 13 lower ranked disorders (those scoring <8.5 points) have a treatment strategy available that changes the prognosis for all forms of the disorder. Looking at all three pillars together, we can see that more than half, 54% (19/35 disorders), meet the following three criteria: (1) “all forms of the disorder are asymptomatic for the first weeks of life”; (2) have a “DBS test available and in use”; and (3) have a treatment strategy where “pre-symptomatic initiation results in better outcomes”.

## 4. Discussion

Currently, in the EU, there are great disparities in the number of disorders screened for between countries. Since 2011, when the European Commission published documentation that supports discussion on how to develop policies for NBS for rare disorders, there has been little concrete progress [6]. The NBS evaluation algorithm [8] could be utilised to objectively assess disorders and build a standard minimal panel of disorders to be recommended for NBS across Europe. Utilising the NBS evaluation algorithm to assess 48 IMDs, there are 35 disorders that most strongly fulfil the Wilson and Jungner classic screening principles and are therefore recommended for inclusion in NBS programmes. Of these 35 top-ranking disorders, all have a rapidly progressing form, 33 have a DBS test available, and 31 have a treatment strategy available.

We need leadership to drive a consistent and methodological approach for NBS programmes across Europe. The majority of the European member states have no legislation governing NBS [5], but certain European countries have their own particular strengths in NBS. In Italy, since 2016, NBS has been legally mandated throughout the country for about 40 IMDs [5]. The United Kingdom national screening committee (UK NSC), established in 1996, has a rigorous evidence review process and meets three times a year to make new recommendations or update existing ones [14]. The Netherlands NBS programmes is very reactive; SCID was added to their NBS panel in under six years [15]. In April 2015, SCID was recommended for inclusion on NBS, in April 2018 the first heel prick blood was collected on 1 April 2018, and on 1 January 2021 SCID was officially added to the national heel prick screening program [15]. In comparison, the timeline for implementing SCID NBS stretched over a 12 year-period in the US, from 2006 when it was nominated for addition to the RUSP, to 2008 when pilots began in Massachusetts and Wisconsin, to 2018 when all 50 states had implemented NBS for SCID. We need to combine the breadth of disorders on the Italian NBS panel, with the rigor of the UK NSC and the agility of the Dutch programme. 

Expanding screening for IMDs at birth has the potential to reduce the time to diagnosis, and the related psychological impact on families and patients, and to allow for early, pre-symptomatic treatment that may change prognosis. NBS programmes also benefit society and the healthcare system because increased patient identification increases our knowledge on natural history, frequency, and genotype/phenotype correlations, and thus can help to advance diagnosis and treatment options [16].

This paper is intended to present a living list of IMDs that, at present, most strongly fulfil objective and measurable clinical criteria, thereby recommending them for further evaluation for inclusion on national NBS programmes. Nevertheless, to accelerate NBS expansion it is critical that countries work together to leverage each other’s success and evidence, especially in the rare disease space because of the severity and rapid progression inherent to so many of these disorders. There are disorders that can be accurately diagnosed via DBS test and have a treatment strategy that can change prognosis if initially pre-symptomatically. We need to expand NBS programmes now to diagnose and treat patients earlier.

## Figures and Tables

**Figure 1 IJNS-08-00020-f001:**
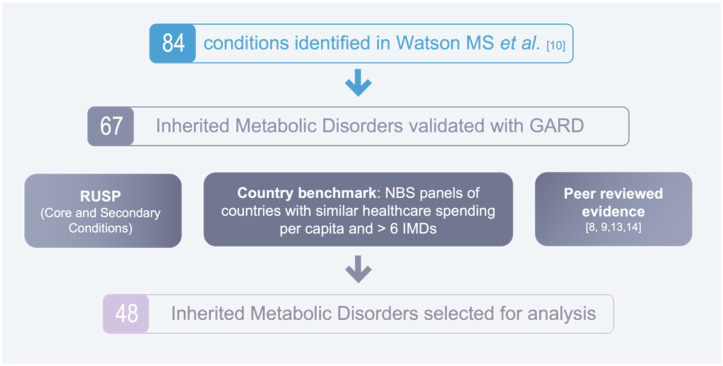
Selection of disorders for analysis [8,9,10,13,14].

**Figure 2 IJNS-08-00020-f002:**
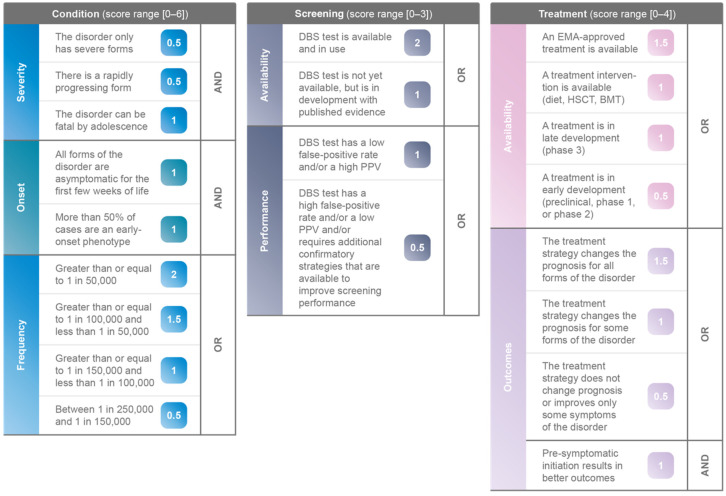
NBS evaluation algorithm [8].

**Table 1 IJNS-08-00020-t001:** Scoring of IMDs using the IMD NBS evaluation algorithm, ranked by highest to lowest score.

Disorder	Score (0–13)	Condition			Screening		Treatment	
		Severity	Onset	Frequency	Availability	Performance	Availability	Outcomes
Carnitine uptake defect/carnitine transport defect (CUD)	12.5	1.5	2	2	2	1	1.5	2.5
Severe combined immunodeficiency (SCID)	12	2	2	2	2	0.5	1.5	2
Glutaric aciduria type 1 (GA1)	11.5	2	2	1.5	2	1	1	2
Homocystinuria (HCU)	11.5	1.5	2	1	2	1	1.5	2.5
Phenylketonuria (PKU)	11.5	0.5	2	2	2	1	1.5	2.5
Tyrosinemia, type 1 (TYR 1)	11.5	1.5	2	1.5	2	0.5	1.5	2.5
Classic galactosaemia (GALT)	11	2	1	2	2	1	1	2
3-Hydroxy-3-methyglutaric aciduria (HMG)	11	1.5	2	1	2	1	1	2.5
Pompe disease	11	1.5	1	2	2	0.5	1.5	2.5
X-linked adrenoleukodystrophy (X-ALD)	10.5	1.5	1	2	2	1	1	2
Argininosuccinic aciduria (ASA)	10.5	2	1	1.5	2	1	1.5	1.5
Carnitine palmitoyltransferase, type I deficiency (CPT I)	10.5	2	2	0	2	1	1	2.5
Long-chain 3 hydroxyacyl-CoA dehydrogenase deficiency (LCHAD)	10.5	2	2	1	2	1	1	1.5
Methylmalonic acidaemia (cobalamin disorders, Cbl A, B)	10.5	2	2	0	2	1	1	2.5
Metachromatic leukodystrophy (MLD)	10.5	2	2	1.5	1	0.5	1.5	2
Mucopolysaccharidosis, type I (MPS I)	10.5	1.5	2	1.5	2	0.5	1.5	1.5
Propionic acidaemia (PROP)	10.5	2	1	0.5	2	1	1.5	2.5
Biotinidase deficiency (BIOT)	10.5	2	1	1.5	2	0.5	1	2.5
Medium-chain acyl-CoA dehydrogenase deficiency (MCADD)	10	1.5	1	2	2	1	1	1.5
3-Methylcrotonyl-CoA carboxylase deficiency (3MCC)	10	1.5	1	2	2	1	1	1.5
Citrullinemia, type I (CIT)	10	1.5	1	0.5	2	1	1.5	2.5
Holocarboxylase synthetase deficiency (MCD)	10	2	1	0.5	2	1	1	2.5
Krabbe disease	10	1.5	2	1.5	2	0.5	1	1.5
Argininaemia (ARG)	9.5	1.5	2	0	2	0.5	1	2.5
Carnitine acylcarnitine translocase deficiency (CACT)	9.5	2	1	0	2	1	1	2.5
Very long-chain acyl-CoA dehydrogenase deficiency (VLCAD)	9.5	1.5	1	2	2	1	1	1
Maple syrup urine disease (MSUD)	9	1.5	0	1	2	1	1	2.5
Methylmalonic acidaemia (methylmalonyl-CoA mutase) (MUT)	9	1.5	1	0.5	2	1	1	2
Carnitine palmitoyltransferase, type II deficiency (CPT II)	9	1.5	1	0	2	1	1	2.5
Batten disease (CLN2)	9	2	2	0	1	1	1.5	1.5
Niemann Pick A/B (ASM deficiency)	9	2	2	0.5	2	1	1	0.5
Isovaleric acidaemia (IVA)	8.5	1.5	0	1	2	1	1	2
Trifunctional protein deficiency (TFP)	8.5	1.5	1	0	2	1	1	2
Gaucher disease	8.5	1.5	1	1.5	2	0.5	1.5	0.5
Lysosomal acid lipase deficiency (LAL-D/Wolman/CESD)	8.5	1.5	1	0.5	2	1	1.5	1
Multiple acyl-CoA dehydrogenase deficiency (MADD)	8	1.5	0	0.5	2	1	1	2
MPS VI (Maroteaux-Lamy syndrome)	8	1.5	2	0	1	0.5	1.5	1.5
Alpha-mannosidosis	7.5	1.5	1	0	1	1	1.5	1.5
Fabry disease	7.5	0	1	1.5	2	0.5	1.5	1
MPS II (Hunter syndrome)	7	0	2	0.5	1	0.5	1.5	1.5
MPS III (Sanfilippo syndrome)	6.5	0	2	1.5	1	1	0.5	0.5
Niemann-Pick type C disease	6.5	1.5	1	1	1	0	1.5	0.5
MPS IV (Morquio syndrome)	5.5	0	2	0	1	0.5	1.5	0.5
Sandhoff disease (GM2 gangliosidosis, type II)	5.5	1.5	2	1	0	0	1	0
Farber disease	5	2	1	0	0	0	1	1
Tay-Sachs disease (GM2 gangliosidosis, type I)	4.5	1.5	2	0	0	0	1	0
MPS VII (Sly syndrome)	3.5	1.5	0	0	0	0	1.5	0.5
MPS IX (hyaluronidase deficiency)	1	0	1	0	0	0	0	0

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
