# Peer review of "Application of a Novel Algorithm for Expanding Newborn Screening for Inherited Metabolic Disorders across Europe"

_2409-515X, 2022, doi:10.3390/ijns8010020_

Round 1

Reviewer 1 Report

The authors intent to attain a unique list of inherited metabolic disorders which can be used in the EU, to avoid disparities across the Europe. For this purpose, they employ a new algorithm, previously developed.

The expansion and the standardization of the screening panel across the Europe is a crucial point for the health of the European babies as well as the attempt to identify a unique panel is a noble aim.

In order to achieve the purpose of the work, the authors refer several times to the need to rely to the published scientific evidence, especially for the choice of the criteria that characterize the “three pillars”. However, all their evaluations are based on an unpublished algorithm. They also stated that “the algorithm was developed based on the Wilson and Jungner criteria”. How the algorithm fit with the updates of the Wilson and Jungner criteria proposed by other authors (see below)?

It would be more correct to resubmit the paper after the reference 2 will be accepted.

Anne Andermann, Ingeborg Blancquaert, Sylvie Beauchampb, Véronique Déryc. Bulletin of the World Health Organization, April 2008, 86 (4)

CMAJ 2018 April 9;190:E422-9. doi: 10.1503/cmaj.171154

Author Response

Review is in black bold, Responses are in blue

The authors intent to attain a unique list of inherited metabolic disorders which can be used in the EU, to avoid disparities across the Europe. For this purpose, they employ a new algorithm, previously developed. The expansion and the standardization of the screening panel across the Europe is a crucial point for the health of the European babies as well as the attempt to identify a unique panel is a noble aim. In order to achieve the purpose of the work, the authors refer several times to the need to rely to the published scientific evidence, especially for the choice of the criteria that characterize the “three pillars”. However, all their evaluations are based on an unpublished algorithm. They also stated that “the algorithm was developed based on the Wilson and Jungner criteria”. How the algorithm fit with the updates of the Wilson and Jungner criteria proposed by other authors (see below)?

The algorithm has considered both the classic and emerging Wilson & Jungner criteria, even though the emerging criteria published in 2008 (Andermann A et al. Bulletin of the World Health Organization. 2008) have not been widely accepted or applied. The tandem publication, IJNS-1502859, has been updated to make this more clear, lines 83-86, as below:

The emerging Wilson & Jungner criteria were also evaluated, however these principles also fall into the category “Other” as they encompass ethical, economic, and societal aspects of screening programmes which are not included in this algorithm [6].

It would be more correct to resubmit the paper after the reference 2 will be accepted.

The manuscripts have been submitted as companion papers to be published sequentially in the same journal.

Reviewer 2 Report

The manuscript by Jones, et al, has the potential to be a hallmark paper utilized by NBS programs worldwide in their efforts to individually assess disease potential for inclusion in NBS programs and harmonize across jurisdictions.

Overall, the paper reads very nicely and the large amount of data are well organized in tables. The minor suggestions presented below only serve to enhance an already well-written and studied paper:

Lines 51-52: While the RUSP's purpose has been to try to standardize NBS across states, this hasn't necessarily been achieved as there continue to be widening gaps in number of diseases screened for by states. Consider acknowledgement of this - that while the RUSP has been put forth with the intention of standardization, the state-based nature of NBS has still resulted in discrepancies.

Line 80-82: It is not clear how the 67 IMDs were selected from the original cross-comparison between the 84 disorders and 505 disorders in GARD. Please elaborate how that list was determined. Likewise, in the second step, please confirm that the final list of 48 IMDs came from that initial list of 67.

Line 81-82: Unclear as to what this sentence means, "...48 IMDs were selected, based on the following three criteria to advantage disorders that are already screened for widely or that have previously been recommended for screening." Is there a typo?

Figure 1 is very difficult to read - a higher quality image is warranted here.

While perhaps not necessary to change for this manuscript, I would lean away from assigning one of the three pillars under the term "Diagnosis." Especially given that the criteria within that pillar address screening assay availability  and performance within a DBS matrix and not the availability of diagnostic centers or testing.

Lines 102-106 - the US has started to rely more and more on so-called "gray" literature/unpublished findings in their evidence assessments. Could you indicate if this type of evidence was at all considered for inclusion in assessing the Condition category?

Lines 137-139 - the count adds up to 45, not 48 for the disorder types. Can you speak to what the other 3 disorders were? This is also true in the 3rd bullet that discusses current countries screening.

Line 202 and 229: Typo fulfil should be fulfill

Conclusion - I found it interesting that some of the higher scoring disorders (at least above the 8.5 threshold) are not screened for anywhere, and are not on the RUSP. Could the authors speak to this? Why do you think these disorders scored so high, and yet, are not being actively screened anywhere?

Author Response

Review is in black bold, Responses are in blue

The manuscript by Jones, et al, has the potential to be a hallmark paper utilized by NBS programs worldwide in their efforts to individually assess disease potential for inclusion in NBS programs and harmonize across jurisdictions.Overall, the paper reads very nicely and the large amount of data are well organized in tables. The minor suggestions presented below only serve to enhance an already well-written and studied paper:

Lines 51-52: While the RUSP's purpose has been to try to standardize NBS across states, this hasn't necessarily been achieved as there continue to be widening gaps in number of diseases screened for by states. Consider acknowledgement of this - that while the RUSP has been put forth with the intention of standardization, the state-based nature of NBS has still resulted in discrepancies.

This is interesting and we believe would also parallel to the application of this algorithm across Europe, as we envision differences at local level (based on critical parts of screening programs that are not considered by the algorithm, e.g. cost-effectiveness).

Line 80-82: It is not clear how the 67 IMDs were selected from the original cross-comparison between the 84 disorders and 505 disorders in GARD. Please elaborate how that list was determined. Likewise, in the second step, please confirm that the final list of 48 IMDs came from that initial list of 67.

We have added a flowchart to explain this process more clearly, see Figure 1. We chose a defined list of disorders to test the algorithm on, but it could be applied more broadly.

Line 81-82: Unclear as to what this sentence means, "...48 IMDs were selected, based on the following three criteria to advantage disorders that are already screened for widely or that have previously been recommended for screening." Is there a typo?

Lines 76-30 have been re-written for clarity and to explain Figure 1.

Figure 1 is very difficult to read - a higher quality image is warranted here.

Agreed, we have replaced this with a higher quality image.

While perhaps not necessary to change for this manuscript, I would lean away from assigning one of the three pillars under the term "Diagnosis." Especially given that the criteria within that pillar address screening assay availability and performance within a DBS matrix and not the availability of diagnostic centers or testing.

Thank you, we agree, we will change the pillar “Diagnosis” to “Screening”. This has also been applied to the tandem publication, IJNS-1502859.

Lines 102-106 - the US has started to rely more and more on so- called "gray" literature/unpublished findings in their evidence assessments. Could you indicate if this type of evidence was at all considered for inclusion in assessing the Condition category?

No, this is built to be evidence-based and we only used medical databases and peer-reviewed literature.

Lines 137-139 - the count adds up to 45, not 48 for the disorder types. Can you speak to what the other 3 disorders were? This is also true in the 3rd bullet that discusses current countries screening.

3 disorders are classified as “Other” (BIOT, Classic galactosaemia and SCID) as they are not LSDs, DFAMs, etc. We have added this information to the manuscript in line 140.

Line 202 and 229: Typo fulfil should be fulfill

Changed, thank you.

Conclusion - I found it interesting that some of the higher scoring disorders (at least above the 8.5 threshold) are not screened for anywhere, and are not on the RUSP. Could the authors speak to this? Why do you think these disorders scored so high, and yet, are not being actively screened anywhere?

Some of these conditions, high scoring but not on RUSP, would be Krabbe and MLD where there have been very recent advances and steps towards newborn screening for these conditions, e.g. NIH ScreenPlus study in New York state, in the past years. The 8.5 threshold was based on the UK NSC as this algorithm was designed for the EU. In this vein that means that we considered EMA-approved treatments (that perhaps have not yet been evaluated by RUSP or the FDA). This is a horizon scanning tool so the intention is that, indeed, there would be high-scoring disorders that are not widely screened for today, but should be considered for addition in the future. The objective is that the algorithm would highlight disorders that merit further evaluation, pilots, cost-effectiveness studies, etc. at the local level.

Round 2

Reviewer 1 Report

The manuscript has been sufficiently improved to warrant publication in IJNS